# Advancements in Attention Decoding Using the MOET Dataset: A Comparative Study

**[Omitted for anonymity.]**

**Editor:** Editor's name

## Abstract

Eye movements serve as a valuable window into understanding the intricacies of the human mind and brain within the field of cognitive science. The analysis of eye movements and gaze fixations has yielded profound insights across various domains, including cognitive science, marketing, human-computer interaction, and human-robot interaction, providing a rich source of knowledge on diverse cognitive functions. A critical challenge in eye-tracking data analysis lies in deciphering a person's visual attention at each moment from their measured gaze behaviour, known as "attention decoding." The majority of eye-tracking data analyses rely on labour-intensive manual coding of attentional states, a slow and error-prone endeavour. Recent advancements in machine learning offer potential automation but were hindered by the lack of publicly available labeled data for benchmarking. The Multiple Object Eye-Tracking (MOET) dataset, a recent release, overcomes this challenge, providing eye-tracking data from human participants observing dynamic visual scenes. We improve upon the existing end-to-end architecture and present several competitive algorithms for the task of attention decoding on the MOET dataset. We also present baseline results for the distinct but related task of labeling the attention loci.

**Keywords:** Gaze; Eye-Tracking; Deep Learning; Attentional Decoding; Visual Attention

## 1. Introduction

Exploring where humans direct their gaze and how it relates to their thinking and actions stands as one of the core inquiries in the realms of psychology and neuroscience (Hayhoe and Ballard, 2005; Koenig et al., 2016). Eye-tracking technology can contribute to understanding what humans look at and how that influences their cognition and behaviour (Franchak et al., 2011; Kurzhals et al., 2016; Bambach et al., 2018). The ability to record and analyze eye movements, as well as gaze fixations, has opened up a world of insights across various fields, including cognition (Kiefer et al., 2017), attention (Kim et al., 2021, 2022), counterfactual simulation (Gerstenberg et al., 2017), offering a rich source of information about diverse cognitive functions. However, a key step of eye-tracking data analysis is determining the locus of a person's visual attention at each point in time. This process is called "attention decoding" since it involves decoding a participant's latent attentional state from their measured gaze behaviour (Uppal et al., 2023). In environments that include only a small number of well-separated objects on a static display, the locus of the participant's attention can be determined using basic automated assessments, such as duration and number of fixations within predefined regions of interest (ROIs) (Carpenter and Just, 1977; Tsai et al., 2012; Raney et al., 2014; Eng et al., 2020; Mirman et al., 2008; Huettig et al., 2011; Kim and Grüter, 2021). However, such an approach fails to imitate the complexity of the real world. Hence, the majority of eye-tracking data analysis is done using

software for hand-coding attentional state frame-by-frame (Steinbach, 1969; Tsang et al., 2010; Kurzhals et al., 2014; Vansteenkiste et al., 2015; Fraser et al., 2017; Miller et al., 2020; Kim et al., 2020; Pellicer-Sánchez, 2016). This is labour-intensive, slow and prone to errors.

While recent tools from machine learning promise to automate this process, a key limitation to the implementation of such tools has been the lack of publicly available labeled data for benchmarking attention decoding. The recently released Multiple Object Eye-Tracking (MOET) dataset (Uppal et al., 2023) overcomes this barrier and provides eye-tracking data collected from human participants viewing several dynamic visual scenes, alongside the corresponding class labels and bounding boxes for the objects which the participants were assigned to track. Previous approaches to tackle this dataset involved the usage of end-to-end deep learning methods and heuristic rule-based methods. The present work builds upon this in three ways: (1) we improve upon existing end-to-end architectures for attention decoding, (2) present competitive alternative algorithms for attention decoding, including heuristic rule-based and two-stage machine learning approaches, and (3) we present the first baseline results for the related task of classifying attention loci on the MOET dataset.

## 2. Related work

**Attention decoding methods:** Typical automated attention decoding methods pass the stimulus video frame-by-frame through an off-the-shelf detector, to obtain candidate predictions for all the objects. These candidate object predictions and the participant's gaze location are used to determine the loci of the participant's attention following deterministic rules. Kumari et al. (2021) and Machado et al. (2019) utilise off-the-shelf pretrained object detectors to identify bounding box of objects in each frame, while Wolf et al. (2018) and Deane et al. (2022) employ segmentation models to obtain segmentation masks. The participant's attention loci is then determined based on which of the candidate objects best explains the gaze locations. Other methods (Panetta et al., 2019; Rong et al., 2022) extract an area around the gaze location and predict the object of interest using either an image classifier or object detector. These methods can perform well when the attentional loci are well-separated but struggle in crowded scenes. Kim et al. (2020) utilise a hidden markov model (HMM) in an artificial visual setting to aggregate temporal information across frames and overcome the challenge of occluded objects.

**Datasets for attention decoding:** Most of the datasets utilised by methods described in the previous section are either not publicly available (Machado et al., 2019; Panetta et al., 2019; Kumari et al., 2021; Deane et al., 2022), limited by size (Wolf et al., 2018), or do not contain the required annotations (Rong et al., 2022) to train attention decoding models. The VISUS dataset (Kurzhals et al., 2014) contains gaze data from 25 participants watching 11 different videos, wherein each participant was assigned a task to perform while watching the video (e.g., follow a red car), with the aim of eliciting particular naturalistic patterns of gaze behaviour. While appropriately annotated for our purposes, the scenes in the VISUS dataset are quite sparse, with most frames containing only a single moving object, making attentional decoding relatively easy on this dataset. The Multiple Object Eye-Tracking (MOET) dataset (Uppal et al., 2023) extended the Multiple Object Tracking 2016 (MOT16) benchmark dataset (Milan et al., 2016) with eye-tracking data obtained from 16 participants viewing 14 videos while tracking distinct target objects through the videos.

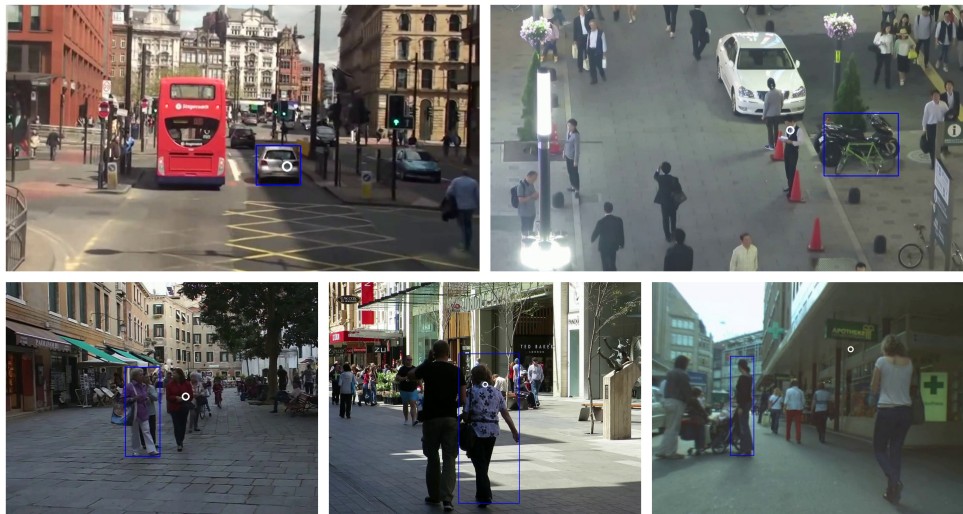

Figure 1: Example frames from the MOET dataset. The blue bounding box indicates the object of interest while the white circle indicates the gaze location.

The MOT16 dataset is a widely used benchmark dataset for multiple object tracking since it provides a large number of crowded moving objects in complex real-world scenes. In the MOET dataset, target objects are annotated in each frame with class labels and bounding boxes. Due to the availability of required annotations and the realistic and challenging nature of MOT16 videos, we utilise the MOET dataset for all our experiments.

## 3. Methodology

### 3.1. Data description and preprocessing

The Multiple Object Eye-Tracking (MOET) dataset extends MOT16 with eye-tracking data obtained from 16 participants viewing the 14 videos while tracking distinct target objects in the videos. Target objects are annotated in each frame with class labels and bounding boxes, with the assigned target object changing roughly 10 times per video. The annotations were obtained from a RetinaNet model (Lin et al., 2017b) trained on the MS COCO dataset.

The experimental setup of the data collection relies on the assumption that the gaze point is always inside the object assigned to be tracked, also known as the overt condition, which is not always the case. The participants track the target imperfectly in some cases as expected. Preliminary analysis revealed that $\sim 42\%$ of the annotated frames do not follow the overt condition; some examples are included in Figure 1. Since the ground truth labels contain a significant amount of noise, data cleaning is crucial before proceeding to training. Similar to Uppal et al. (2023), we perform the following data-cleaning steps:

- **Imputation of missing data:** The gaze position is interpolated linearly from adjacent non-missing frames whenever we encounter the gaze data missing for short sequences ($< 10$ frames, $\approx 167\,\text{ms}$), which typically reflect blinks. Remaining frames with missing gaze, or in which no objects were detected, are omitted from the dataset.

- **Verifying overt condition:** Since the participant's gaze is not always directed towards the specified target object (either because the participant is off-task or because they are transitioning between two targets), Uppal et al. (2023) apply a distance threshold, dropping frames from the training data in which the gaze is more than a certain Euclidean distance (100 pixels) away from the nearest point of the bounding box of the assigned target object. As discussed in Section 4.2, we experiment with several different values of this threshold using their original end-to-end architecture.

For evaluation, we split the dataset across participants since splitting across videos leads to the data being in different domains. We also note that participants 2 and 9 were excluded from the evaluation due to duplicate and incomplete gaze data, respectively. For each video, we train a model on 80% of the participants (11 participants) and then evaluate the performance on the remaining 20% of the participants (3 participants). This emulates applications in which we have labeled data from some participants watching a video and want to apply the model to data from new participants watching the same video, that is, generalization across participants. We measure performance by the mean (across frames) Intersection over Union (IoU) between the predicted and ground truth bounding boxes.

### 3.2. End-to-end attention decoding architectures

**Architecture overview:** Uppal et al. (2023) proposed an end-to-end attention decoding model (Figure 2), which uses a pretrained object detector backbone and incorporates gaze using gaze density maps. The model consists of a ResNet-50 Feature Pyramid Network (FPN; Lin et al., 2017a) backbone extracted from a Faster R-CNN model (Ren et al., 2015), pretrained on the "Common Objects in COntext" (COCO) dataset (Lin et al., 2014).

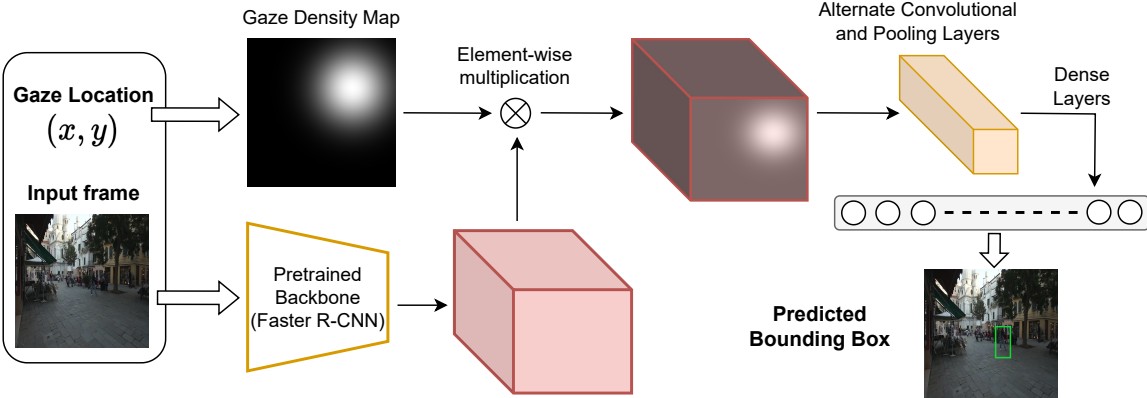

Figure 2: End-to-end Model Architecture

For each input frame $I$, the highest-resolution (128x128) feature map $F(I)$ is obtained from the FPN. Each input frame $I$ is associated with a gaze point $g = (x, y)$. In order to capture the spatial attention of the participant, the gaze point $g$ is transformed into a gaze density map $\text{GDM}(g)$, which is represented by a Gaussian, centred at the gaze location:

$$\text{GDM}(g) = \mathcal{N}(g, \sigma), \tag{1}$$

where the standard deviation $\sigma$ is a model hyperparameter. This can be interpreted as a feature importance map, describing the spatial attention of the participant at the current frame. We combine $F(I)$ and $GDM(g)$ in a weighting mechanism, similar to the approach proposed by Sattar et al. (2020). Specifically, $GDM(g)$ is downsampled to match the resolution of $F(I)$ and is integrated as shown below:

$$F_{\text{wgt}}(I) = F(I) \otimes GDM(G) \tag{2}$$

where $\otimes$ stands for element-wise multiplication. The weighted feature map $F_{\text{wgt}}(I)$ is passed through 2 alternating 3x3 convolutional and 2x2 pooling layers, doubling the depth at each step. Finally, we pass it through three dense layers to predict the final bounding box. Note that Uppal et al. (2023) explored using both GRU units and dense layers at the end to gauge the effect of temporal information to combat object collisions and occlusions. Since they concluded that temporal information propagated by the GRU units did not improve model performance, we only take into account a single frame at a time.

**Training procedure:** The weights of the pretrained ResNet-50 FPN backbone network are frozen during training. Each model is trained for 150,000 iterations, using $L_1$ loss and Adam for stochastic optimization, with learning rate $10^{-3}$ and weight decay $10^{-4}$. We keep $\sigma = 500\,\text{px}$, following Uppal et al. (2023). We note that the training of this model is quite unstable and hence layer normalization is added in the dense layers. Apart from this, thresholding to ensure the overt condition (see Section 3.1) is also omitted. As discussed in Section 4, these two changes to the training procedure lead to significant performance improvements over the procedure of Uppal et al. (2023).

### 3.3. Two-Stage method

We also present a simple two-stage method using a pretrained object detector, as illustrated in Figure 3. Faster R-CNN, pretrained on the MS COCO dataset, is used to obtain the top-$k$ bounding boxes (for some predetermined model hyperparameter $k$), based on their predicted probability, for each frame. All the bounding boxes are concatenated into a single vector, alongside the corresponding gaze point for that frame. The collection of such vectors is used to train a multi-output random forest regressor with 1000 estimators which predicts the bounding box of the object of interest. In the case where $k$ is greater than the number of objects predicted by Faster R-CNN, the remaining elements of the vector are set as 0.

### 3.4. Heuristic baselines

We include the following heuristic baselines proposed by Uppal et al. (2023):

- **Fixed Box Baseline:** For each video, the fixed-box baseline predicts a bounding box of fixed size centred around the gaze point. The size of the bounding box is the average size of all bounding boxes, across all training participants, for that video.

- **Object Detector (OD) baseline:** Using the Faster R-CNN model, pretrained on the MS COCO dataset, candidate bounding box predictions are generated for each input frame. If the gaze location is present inside any of the candidate bounding boxes, it is chosen to be the object of interest. In the case of overlapping bounding

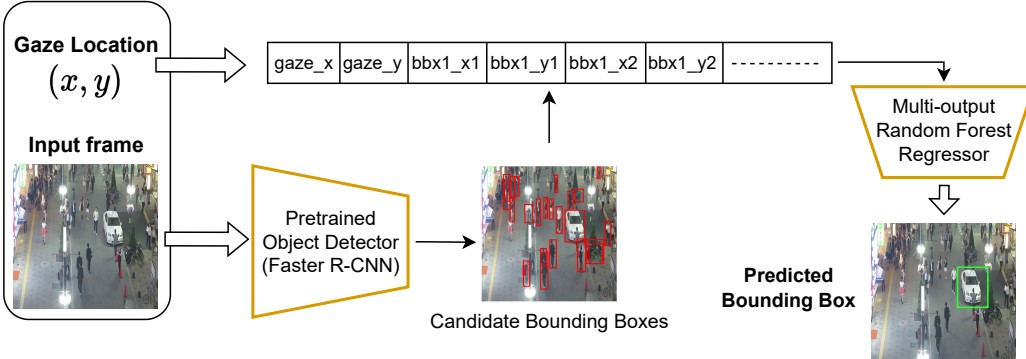

Figure 3: Two-Stage Model Architecture

boxes, the one with the highest prediction probability (as given by the Faster R-CNN model) is chosen. If the gaze location does not lie inside any of the predicted bounding boxes, then a random candidate is chosen to be the object of interest.

- **Object Detector (OD) mod:** It is similar to the above baseline with a slight modification in the case when the gaze location does not lie inside any of the candidate bounding boxes; in such a scenario, the Euclidean distance between the gaze point and the nearest point of all the candidate-bounded boxes is calculated, and the one with the least value is chosen as the predicted bounding box.

- **Object Detector (OD) oracle:** The "OD Oracle" model is given access to the true target bounding box and selects the best bounding box out of those provided by the object detector; while this is not feasible in practice, the performance of this model provides a reference upper bound on the best performance possible for any model selecting one of the bounding boxes output an object detector.

## 4. Experimental Results

All the experiments were performed on Ubuntu 20.04.2 LTS, with implementation using PyTorch 1.11 and logging using the Weights & Biases (`wandb`) toolbox. Code for reproducing our results is available at [Omitted for anonymity.].

### 4.1. Generalisation across participants

Averaged results across all the 14 videos suggest that the best results are achieved by the OD Mod method, which applies heuristics to combine gaze with the output of an off-the-shelf object detector, followed closely by the Two-Stage model, as seen in Table 1. Since the performance of the end-to-end model is lower than that of the OD Mod baseline, both of which are much lower than that of the OD Oracle with access to the original bounding boxes, we speculate that the overall error is dominated by errors in the initial estimation of the bounding boxes, rather in the downstream selection of the best bounding box.

| Approach | Algorithm | Mean IoU (Std. Dev.) |
|---|---|---|
| Completely Rule-Based | Fixed Box Baseline | 0.2203 (0.0713) |
| Object Detector (OD) + Rule | OD Baseline | 0.4769 (0.1177) |
| | OD Mod | 0.5091 (0.0990) |
| | OD Oracle | 0.7088 (0.0740) |
| Machine Learning | Uppal et al. (2023) | 0.3593 (0.1214) |
| | End-to-end Method (Ours) | 0.4358 (0.1164) |
| | Two-Stage Method (Ours) | 0.4941 (0.1290) |

Table 1: Means and standard deviations (across test set participants) of mean IoUs (across 14 videos), for each algorithm. Faded values indicate cases where the model uses "oracle" knowledge of the true bounding box and are provided only for comparison.

## 4.2. Hyperparameter Tuning

**Distance threshold for end-to-end method:** Uppal et al. (2023) apply a crucial data-cleaning step of verifying the overt condition. We experiment on different values of this threshold (50, 100, 200, 500 and no threshold) on their end-to-end attention decoding architecture, trained and evaluated for all the participants on 7 of the 14 videos. As seen in Table 2, the model performance is greatest when trained without any training threshold. Hence, we use no threshold when training our end-to-end attention decoding models.

| Training Threshold | 50 | 100 | 200 | 500 | None |
|---|---|---|---|---|---|
| Mean IoU | 0.4094 | 0.3971 | 0.4061 | 0.3946 | 0.4264 |

Table 2: Mean IoUs (across 7 of the 14 videos, for all the participants) for the end-to-end architecture proposed by Uppal et al. (2023).

**Number of bounding boxes and distance threshold for Two-Stage method:** As above, experimented with the distance threshold, as well as the parameter $k \in \{10, 20, 50, 100\}$, for the Two-Stage method; results are in Table 3. In contrast to the end-to-end model, the Two-Stage Method performed best with a distance threshold of 50 and $k = 10$.

| No. of top detections, $k$ | Distance Threshold | | | | | |
|---|---|---|---|---|---|---|
| | 0 | 50 | 100 | 200 | 500 | None |
| 10 | 0.4817 | 0.4941 | 0.4904 | 0.4724 | 0.4422 | 0.3969 |
| 20 | 0.4799 | 0.4931 | 0.4899 | 0.4731 | 0.4421 | 0.3978 |
| 50 | 0.4789 | 0.4929 | 0.4905 | 0.4742 | 0.4455 | 0.4008 |
| 100 | 0.4790 | 0.4928 | 0.4912 | 0.4747 | 0.4460 | 0.4016 |

Table 3: Mean IoUs (across all 14 videos) for test set participants for the Two-Stage Method for varying values of $k$ and distance threshold for dataset cleaning.

| Models | Accuracy | Weighted $F_1$ Score |
|---|---|---|
| OD Baseline | 85.84 % | 0.8877 |
| OD Mod | 89.11 % | 0.9102 |
| OD Oracle | 94.10 % | 0.9565 |
| Two-Stage Method | 88.50 % | 0.8742 |

Table 4: Classification metrics for test set participants (across all videos) for the task of labeling attention loci. Faded values indicate cases where the model uses "oracle" knowledge of the true bounding box and are provided only for comparison.

### 4.3. Classifying Attention Loci

Whereas our previous experiments focused on estimating a bounding box for the locus of the participant's attention, in many applications, it is useful to explicitly classify the object the participant is attending to. To the best of our knowledge, there are no existing comparable benchmarks for this latter task; here, we present the first baselines for this task using our two-stage models. The dataset primarily contains annotations for 5 classes: person (86%), car (6%), motorcycle (1.7%), chair (1.7%), bus (1.5%), and other miscellaneous classes ($<$ 1%). Due to this class imbalance, we report both accuracy and weighted $F_1$ score. For OD Baseline, OD Mod and OD Oracle, the predicted class label for the corresponding bounding box predicted by the pretrained object detector is considered the attention locus label. For the Two-Stage method, the class label for the candidate bounding box having maximum IoU with the predicted bounding box is taken as the predicted label. As shown in Table 4, OD Mod performed best in terms of both accuracy and $F_1$ score, although the Two-Stage Method gives competitive performance terms of accuracy.

## 5. Conclusion

Attention decoding is a challenging computer vision task with limited prior literature. The Multiple Object Eye-Tracking (MOET) dataset provides a benchmark for the development of automated attention decoding algorithms. This paper presents a comprehensive study on tackling the problem of attention decoding on the MOET dataset and provides several competitive baselines, including heuristic rule-based methods, two-stage and end-to-end machine learning methods. We also improve upon the end-to-end architecture proposed by Uppal et al. (2023) and present baseline models for the task of labeling attention loci.

Our results highlight the challenge of developing end-to-end machine learning models that convincingly outperform simpler heuristics; specifically, end-to-end models were generally outperformed by the two-stage methods, combining pretrained object detectors with heuristic rules or random forest models. We note that the performance of all methods on the MOET dataset may be limited by artefacts in the construction of the original data, in particular the idiosyncratic method that was used to construct the "ground truth" bounding boxes during data collection. This would help explain the substantial gap between the performance of the "oracle" algorithm and the other approaches, and why preprocessing the data with different distance thresholds considerably changes performance.

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
