# OpenReview forum: "Advancements in Attention Decoding Using the MOET Dataset: A Comparative Study"
_NeurIPS.cc/2023/Workshop/Gaze_Meets_ML — Submitted to Gaze Meets ML 2023_

### Official Review · Reviewer_sE4F · 2023-10-21
**Comparing attention decoding methods on new MOET dataset**

**Rating:** 7
**Confidence:** 3

**Review:**

This paper presents an extended analysis of attention decoding using the newly released MOET dataset. Attention decoding is the task of localizing a subject’s center of attention during a video based on their gaze positions and saccades. The authors claim to make three contributions. First, improving previous machine learning models for attention decoding. Second, defining a couple of new two-stage model architectures and comparing them with existing machine learning and heuristics approaches. Finally, presenting the first results for classifying the subjects of attention loci on MOET. Overall, although it does not explore ambitious or very novel methods, this paper is a thorough comparative study and adds to the growing attention decoding literature. However, I am not familiar with this literature, so I may not be the most qualified to judge the significance of this comparative study.

There are several things I like about this paper:
* The paper is clearly written and well-organized.
* Attention decoding is an important problem to study. Even in cognitive science, the boundaries between eye gaze and latent attention are sometimes blurred, so explicitly defining the problem of decoding attention from eye movements is valuable.
* The authors provide good justification for the data cleaning steps and each of the heuristic baselines.
* The model architecture figures are clear, and the results tables are clear as well.

My critiques and suggestions are as follows:
* As a general critique, the framing of the study seems more to summarize previous literature and provide a jumping-off-point for future studies, rather than contribute a large jump itself. Indeed, the data cleaning, end-to-end architecture, and baselines were all taken from Uppal et al. (2023). A comparative study is also valuable, but it does detract from the novelty of methods.
* Gaze fixation does not coincide with latent attention, yet this is the implicit assumption that is made when you center the “attention” Gaussian at the gaze fixation in the end-to-end model. I can easily look at one thing yet focus my attention on another. The authors simply used the same architecture as Uppal et al. (2023), but it would be good to address this point.
* The two hyperparameter tuning subsections show test results, but do not provide any explanation to the reader about why this might be. For instance, why do layer normalization and removing distance threshold help the end-to-end model? Is there any intuition on why not having a distance threshold works best for the end-to-end method, yet not for the two-stage method? I am particularly curious if these observed hyperparameter trends are interpretable/explainable enough that they could be expected to generalize across model architectures and even different datasets, which would make such hyperparameter searches much more scientifically significant. This section could add a lot more to the paper if questions of this type were addressed.
* It is suggested more than once that the error may be dominated by errors in the initial estimation of bounding boxes. Given this observation, what may be the next steps for better attention decoding performance on MOET?
* I would consider placing Table 4 after the text in 4.3.

---

### Official Review · Reviewer_EfNH · 2023-10-21
**Limited technical contribution - Mainly built upon a previous study with minor improvements - nice comparison tables on MOET data**

**Rating:** 6
**Confidence:** 4

**Review:**

This study proposes a pipeline for the analysis of eye-tracking data with focus on attention decoding, a process to understand a person`s visual attention patterns over time. The algorithm integrates a pre-trained object detector with gaze density maps as attention modelling. The study is developed on MOET dataset.

This work extends upon prior research conducted by Uppal et al., by enhancing the existing end-to-end architecture by introducing normalization layers and excluding the threshold for overt conditions. While these modifications optimize the training process, they are relatively minor technical contribution when compared to Uppal et al.`s foundational work.

Nevertheless, the study is well-written, well-conducted and offering a comparison of various algorithms performance on the MOET database, establishing a baseline for future studies. Therefore, the study could be accepted.

There is a minor issue in Equation 2 that should be corrected for clarity (GDM(g))

---

### Official Review · Reviewer_67qh · 2023-10-24
**Improved benchmarks on a dataset that has inaccurate ground truth & fundamental assumptions**

**Rating:** 3
**Confidence:** 5

**Review:**

The authors are presenting two new methods/models to decode attention in a video + gaze dataset.
There are a few fundamental issues with this line of work which unfortunately goes beyond this manuscript. First, narrowing down the mechanism of attention to the user's gaze point falling in the bounding box of a certain object in the scene is too simplistic and in fact in complete disagreement with how humans attend to a scene. Second, the task, as has been shown in numerous studies does play a significant role in defining the user's attention mechanism and strategies. Third, in many studies, it is shown that even if the users are strictly instructed to follow a certain object in the scene, their gaze will fall behind or ahead of the target, and furthermore, it doesn't always fall on the so-called fovea. In fact, in many time-sensitive tasks, it is shown that for target interception (which is a much more reasonable task to ask a user to perform compared to just following a person in a video) humans tend to NOT track the target over the entire trajectory. So, in essence, I see a fundamental problem in the basic assumptions of the paper, especially in the introduction section, where this work is strongly tied to investigating attention. I suggest revising the motive to look into the oculomotor behavior of users while being asked to track a certain object in the scene.

Another major problem is, we don't really know by the end of the paper why the 30% missed instances in the dataset could not be inferred correctly. The reader is informed towards the beginning of the paper that the dataset has issues with gaze points not always falling onto an object-bounding box, which exactly reiterates my previous point about the expectation that the human gaze being a passive tracking system is flawed. While it is not surprising that the dataset has serious problems in terms of data cleaning and vetting, it would be helpful to know how many instances of which type of problem had to be removed. (i.e. 42% of the dataset did not conform with its own assumption!?)

No comparison of crowded vs less cluttered scenes was presented in the results section, even though it seems that the authors are aiming to tackle this problem as well.

The assumption of training on a set of users and testing on others is not very well supported. In that, two different users might have two different oculomotor behaviors in the sense that their gaze tracking, fixations, and smooth-pursuit patterns would most likely be different. Wouldn't it be better to mix between users in order to capture the between users' variability?

The results in the table for parameter tuning seem almost equal in many scenarios, wouldn't be helpful to run a statistical test to differentiate better? This appears as a bias or preference on a certain parameter set that are not strongly backed by numbers.

If the gaze doesn't fall on a bounding box, why would the authors expect a random selection could perform better? In other words, how would this inform our underlying understanding of attention?

"We speculate that the overall error is dominated by errors in the initial estimation of the bounding boxes, rather in the downstream selection of the best bounding box." Could you elaborate on this? What is the basis of this speculation?

The new benchmark is not very well explained and only in one paragraph it is hard to understand the class labels. For instance, authors mentioned that "person (86%), car (6%), motorcycle (1.7%), chair (1.7%), bus (1.5%)," But how does chair appear among other classes that are seemingly more relevant to each other? Is this a mistake? chair would make sense to be in a dataset with indoor furniture.

---

### Meta-Review · Area_Chair_jyg8 · 2023-10-26

**Recommendation:** Reject
**Confidence:** 4

**Metareview:**

The paper tackles the intricate challenge of 'attention decoding' in eye-tracking data analysis. While the reviewers commend the paper's organization and clarity, they have identified several concerns that require attention. I recommend enhancing the paper by addressing these reviewer concerns.

---

### Decision · Program_Chairs · 2023-10-26

Reject